# A Comprehensive Review of Risk Factors for Venous Thromboembolism: From Epidemiology to Pathophysiology

**DOI:** 10.3390/ijms24043169

**Published:** 2023-02-05

**Authors:** Daniele Pastori, Vito Maria Cormaci, Silvia Marucci, Giovanni Franchino, Francesco Del Sole, Alessandro Capozza, Alessia Fallarino, Chiara Corso, Emanuele Valeriani, Danilo Menichelli, Pasquale Pignatelli

**Affiliations:** 1Department of Clinical Internal, Anesthesiological and Cardiovascular Sciences, Sapienza University of Rome, Viale del Policlinico 155, 00161 Rome, Italy; 2Department of General Surgery and Surgical Specialty Paride Stefanini, Sapienza University of Rome, 00161 Rome, Italy

**Keywords:** venous thromboembolism, deep vein thrombosis, pulmonary embolism, SARS-CoV-2, cancer, antiphospholipid syndrome, thrombophilia

## Abstract

Venous thromboembolism (VTE) is the third most common cause of death worldwide. The incidence of VTE varies according to different countries, ranging from 1–2 per 1000 person-years in Western Countries, while it is lower in Eastern Countries (<1 per 1000 person-years). Many risk factors have been identified in patients developing VTE, but the relative contribution of each risk factor to thrombotic risk, as well as pathogenetic mechanisms, have not been fully described. Herewith, we provide a comprehensive review of the most common risk factors for VTE, including male sex, diabetes, obesity, smoking, Factor V Leiden, Prothrombin G20210A Gene Mutation, Plasminogen Activator Inhibitor-1, oral contraceptives and hormonal replacement, long-haul flight, residual venous thrombosis, severe acute respiratory syndrome coronavirus 2 (SARS-CoV-2) infection, trauma and fractures, pregnancy, immobilization, antiphospholipid syndrome, surgery and cancer. Regarding the latter, the incidence of VTE seems highest in pancreatic, liver and non-small cells lung cancer (>70 per 1000 person-years) and lowest in breast, melanoma and prostate cancer (<20 per 1000 person-years). In this comprehensive review, we summarized the prevalence of different risk factors for VTE and the potential molecular mechanisms/pathogenetic mediators leading to VTE.

## 1. Introduction

Venous thromboembolism (VTE), which includes deep vein thrombosis (DVT) and/or pulmonary embolism (PE), is associated with an increased global burden and represents the third most common cause of death worldwide [1].

The incidence rate for VTE in a previous study on the general population in Norway was estimated at 1.43 per 1000 person-years, while that for DVT was 0.93 per 1000 person-years and that for PE was 0.50 per 1000 person-years [2]. The estimated incidence of VTE according to different countries is reported in Table 1.

However, during the last decades, several improvements in diagnostic and therapeutic management yielded a linear decrease in VTE-related incidence and deaths from 12.8 to 6.5 per 100,000 persons, without substantial sex-specific differences [14].

The healthcare costs attributable to VTE are estimated at EUR 1.5–2.2 billion regarding annual hospitalisations in Europe [15] and USD 7–10 billion in the United States of America, with USD 12,000–14,000 in the first year for each survivor of a VTE event [16].

As VTE is a preventable disease in many cases, an early risk stratification of patients with the identification of high-risk patients may lead to more effective therapeutic strategies, with an estimated saving of EUR 0.5–1.1 billion/year [14,15].

Indeed, despite some VTE events occurring without any apparent major reason (the so-called “unprovoked” VTE), most VTEs share one or more identifiable risk factors that may cause or facilitate the occurrence of VTE. These may be related to patients’ characteristics, usually permanent, as well as to acute clinical conditions, usually transient [17]. Furthermore, there is convincing evidence that the risk of VTE increases along with the number of predisposing factors [17]. Not all identified factors give the same risk for VTE and, based on the results of large observational studies, they were broadly classified into weak (defined as an odds ratio <2), moderate (defined as an odds ratio 2–9) and strong (defined as an odds ratio >9) by the 2019 ESC Guidelines for the diagnosis and management of acute pulmonary embolism developed in collaboration with the European Respiratory Society (ERS) [18] (Table 2). While strong risk factors, such as major surgery, trauma or hip fracture, are generally actively searched by clinicians who adopt effective VTE prophylaxis strategies, the presence of a moderate or weak risk factor may be unrecognized, requiring a careful clinical evaluation [18]. Furthermore, the exact risk of patients concomitantly presenting with more than one weak or moderate risk factor is yet to be established. Several risk stratification models have been proposed to help clinical judgement, but consensus about the best one to use is still lacking. For instance, in hospitalized medical patients, the most widely evaluated were the Caprini, the Padua and the IMPROVE scores [19]. Conversely, the Khorana, Vienna, PROTECHT and CONKO scores may be used to predict the risk of VTE in cancer patients [20]. However, not all these scores received a convincing validation, so that a careful individual evaluation of risk factors is needed.

Compared to previous works on this topic [17], the aim of this comprehensive narrative review is to provide a general updated overview on the epidemiology of old and new risk factors for VTE considering the advances in the field of thrombosis that occurred in the last decade, mainly related to cancer, antiphospholipid syndrome and, more recently, COVID-19, and to describe the pathophysiological mechanisms through which each factor contributes to the risk of VTE.

## 2. General Mechanisms of Venous Thromboembolism

VTE is characterized by the development of thrombi in virtually all venous districts. Venous thrombi are more frequently found in valve pockets and dilated sinuses of the lower limbs, and they have a laminar structure enriched with platelets, red blood cells, leukocytes and fibrin [21]. In the first phases of thrombus growth, they are not closely adherent to the endothelium, and they may lead to PE [22].

Virchow’s triad has historically represented a simple method to summarize the pathophysiology of VTE, including blood flow stasis, hypercoagulability, endothelial dysfunction and injury [23]. This schematic representation is still valid, and the most commonly identified risk factors of VTE play a role in at least one element of Virchow’s Triad. Table 3 summarizes the mechanism of action of common VTE risk factors. However, recurrent VTE is often idiopathic with no clear detectable risk factor [24].

The pathogenesis of VTE is, however, complex, and other mechanisms, including endothelial damage, macrophages, red blood cells and platelets (PLTs), have been investigated [25].

The first hit in the development of VTE is related to an injury of the venous endothelium due to local/systemic inflammation and blood hypoxemia. Indeed, an unharmed and intact vein vessel expresses thrombomodulin, endothelial protein C receptor, tissue factor (TF) pathway inhibitor and heparin-like proteoglycans that explicate an anticoagulant effect [26,27] and ectonucleotidase CD39/NTPDase1, nitric oxide (NO) and prostacyclin that contribute to endothelial function and vasodilatation and inhibition of PLTs aggregation [27,28,29]. However, an endothelium injury leads to a downregulation of the above-mentioned anticoagulant pathways and to an upregulation of prothrombotic proteins, such as Tissue Factor (TF) [27], and adhesion molecules, such as P-selectin, E-selectin and Von Willebrand factor (vWF) that trap leukocytes and PLTs (Figure 1) [27,30,31]. The expression of prothrombotic molecules, such as P- and E- selectin and vWF, seems to be increased by low blood flow [22,27].

Afterwards, PLTs and white blood cells (WBCs) bind vWF and P- and E- selectin, and WBCs start expressing TF, triggering coagulation cascade activation [21,22]. In addition, among WBCs, polymorphonuclear neutrophils (PMNs) release neutrophil extracellular traps (NETs). These are constituted by extracellular desoxyribonucleic acid (DNA), histones and neutrophil antimicrobial proteins and may contribute as a scaffold to thrombus formation [21,32], inducing platelets adhesion and activation and maintaining the thrombus stability like fibrinogen and von Willebrand factor [33].

Several factors modulate PLT activation, such as injury and inflammation [34]; indeed, PLTs express intracellular and membrane toll-like receptors (TLRs) that bind Damage-Associated Molecular Patterns (DAMPs) and Pathogen Associated Molecular Patterns (PAMPs), resulting in PLTs activation. PAMPs and DAMPs are also recognized by TLRs expressed by various immune cells [35,36,37]. After binding PAMPs and DAMPs, immune cells generate reactive oxygen (ROS) and nitrogen species (NOS), and produce cytokines, such as tumour necrosis factor α, interleukin 8 and interferon (Figure 1) [37,38].

Therefore, the binding of platelets to vWF and WBCs to P-selectin and E-selectin lead, on the one hand, to platelet activation and aggregation and, on the other, to the release of TF. In particular, the increased production of TF leads to the activation of the extrinsic pathway of coagulation, with final formation of fibrin and entrapment of red blood cells and PLTs, resulting in a red-blood-cells-rich thrombus formation [22,27].

## 3. Risk Factors for Venous Thromboembolism

### 3.1. Male Sex

Growing evidence suggests that male sex represents a risk factor for VTE. In a work performed by Baglin et al., VTE recurrence after a first unprovoked event was 25.7% in men and 11.7% in women after 2 years follow-up [39]. A similar difference was observed by Kyrle et al., who registered recurrence rates of 20% and 6%, respectively, at 26 months follow-up, with an age-adjusted RR of 3.6 for male sex [40]. At 5 years, the likelihood of recurrence was 30.7 percent among men, as compared with 8.5 percent among women (*p* < 0.001) [40]. Data from the PROLONG study showed how VTE recurrence was higher in men than women (7.4% vs. 4.3% patient-years, Hazard Ratio [HR]: 1.7, *p* = 0.027) [41].

A patient-level meta-analysis of 7 prospective studies with 2554 patients with a first VTE showed that recurrent VTE at 1 year was higher in men than women, both at 1 year (9.5% vs. 5.3%) and at 3 years (11.3% vs. 7.3%) [42]. The risk of an unprovoked first VTE was higher in men (HR 2.2, 95%CI, 1.7–2.8) also after adjustment for hormone-associated VTE (HR 1.8; 95% confidence interval [95%CI] 1.4–2.5) [42]. However, the risk of provoked VTE following a major risk factor was similar between men and women. In particular, in women, the risk of recurrent VTE was lower when the initial event was related to hormonal causes compared to those without previous hormone use (HR 0.5, 95%CI, 0.3–0.8) [42].

An analysis of 4 European cohorts (CARROT study; CVTE study; AUREC study; and LETS follow-up study), including 2185 patients with a first VTE (1043 men and 1142 women) showed that men had a 2.8-fold higher risk of recurrent VTE than women [43]. This risk was as high as 5.2-fold in men than in women with reproductive risk factors, and 2.3-fold (95%CI, 1.7–3.2) higher in men than in women without reproductive risk factors [43].

However, the factors behind these divergent results are still unclear. Olié et al. found that risk factors for VTE recurrence may differ between male and female: Factor V Leiden mutation was a risk factor for recurrent VTE in male patients (HR 3.5, 95%CI, 1.5–8.1), while in women, the VTE recurrence rate seems to be more related to age at first event (HR 1.3, 95%CI, 1.1–1.5) and obesity (HR 2.5, 95%CI, 1.1–5.5) [44]. A risk modifier may be represented by aging. In a previous study, the incidence of DVT remained constant for males across all age strata, while it was lower for females <55 years and increased for those >60 years [45].

A recent study including 13,932 men with VTE, those aged <50 and ≥50 years without risk factors, had a recurrence risk of 10% at 2 years [46]. For men with risk factors aged <50 years, this risk ranged from 6% (major surgery) to 16% (history of cancer), and in those ≥50 years, from 7% (major surgery) to 12% (ischemic heart disease, chronic obstructive pulmonary disease, and chronic renal disease) [46].

### 3.2. Diabetes

Most epidemiological studies demonstrate an increased risk of VTE among diabetic patients. A metanalysis with 803,627,121 participants and 10,429,227 VTE patients showed that diabetes was associated with a higher risk of VTE (HR, 1.35; 95%CI, 1.17–1.55) [47]. Furthermore, patients with diabetes who developed VTE are more likely to suffer a complicated clinical course and to suffer from recurrent DVT (14.9% vs. 10.7%) and long-term major bleeding complications (16.4% vs. 11.7%) [48].

Despite this convincing evidence, it is still unclear if poor glycaemic control may have a role in increasing the risk of VTE. In a case control study performed on 188 patients with VTE and 370 controls, increased glucose levels measured at presentation were associated with VTE [49]. Thus, compared to the first quartile (blood glucose < 5.3 mmol/L), the third (5.7–6.6 mmol/L) and fourth quartile (≥6.6 mmol/L) were associated with an increased risk of DVT (Odds Ratio [OR] 2.04, 95%CI, 1.15–3.62 and OR 2.21, 95%CI, 1.20–4.05, respectively) [49].

Data from the “Atherosclerosis Risk in Communities” (ARIC) study performed on 12,298 patients showed that haemoglobin A1c levels were not associated with VTE [50]. Similar findings were reported in the prospective cohort of “Swiss Cohort of Elderly Patients with Venous Thromboembolism” (SWITCO 65+), which enrolled 888 elderly patients with acute VTE [51].

On the other hand, a case-control study performed on 2653 VTE patients and 10,612 controls showed that female patients with diabetes and HbA1c levels >7% may have a slightly higher risk for VTE than those with HbA1c 6.5–7.0% [52].

It has been, therefore, questioned whether the increased risk of VTE associated with diabetes may result from the presence of confounders, such as concomitant comorbidities, rather than to diabetes itself [53]. Indeed, concomitant cardio-metabolic disorders may contribute to the hypercoagulability and endothelial dysfunction described in diabetic patients, such as decreased levels of protein C and increased levels of tissue factor (TF), fibrinogen and coagulation factors VII, VIII and XIII [54,55]. A recent study detected increased thrombin generation, along with a higher number of circulating microparticles bearing endogenous pro-coagulant triggers in plasma samples of diabetic (type II) patients [56].

### 3.3. Smoking

Smoking is a well-proved risk factor for atherosclerotic disease, but its association with VTE is less established. A metanalysis that included participants without cardiovascular disease or VTE at baseline showed that current smoking was associated with an increased VTE risk (HR 1.38 95%CI, 1.20–1.58). The HR remained similar after excluding patients diagnosed with cancer [57], that is, in contrast with previous studies linking excess VTE risk to an increased hospitalization for smoking-related diseases, including cancer [58,59]. A more recent metanalysis has shown that current smoking was significantly associated with VTE prevalence in case-control studies (OR 1.34, 95%CI, 1.01–1.77), but not in cohort studies (Relative Risk [RR] 1.29, 95%CI, 0.96–1.72) [60].

The association between smoking and VTE risk seems to be dose-dependent. A large metanalysis showed that compared with never smokers, the overall combined relative risks (RRs) for developing VTE were 1.17 (95%CI 1.09–1.25) for ever smokers, 1.23 (95%CI 1.14–1.33) for current smokers and 1.10 (95%CI 1.03–1.17) for former smokers [61]. The risk increased by 10.2% (95%CI 8.6%–11.8%) for every additional 10 cigarettes smoked per day or by 6.1% (95%CI 3.8%–8.5%) for every additional 10 packs per year [61].

Nicotine and other addictive substances in cigarettes have been shown to increase the percentage of ROS, determining reduced nitric oxide (NO) availability and generating an inflammatory and prothrombotic microenvironment [62]. The loss of the NO-related protective effect and increased ROS production increase platelet reactivity and lipid peroxidation.

### 3.4. Obesity

Obesity is a well-known risk factor for cardiovascular disease, but it is considered a weak risk factor for VTE [18]. In the Tromso study including 6170 subjects aged 25–84 years, abdominal obesity was the only component of the metabolic syndrome associated with VTE (HR 2.03, 95%CI, 1.49–2.75) [63].

In an analysis from the Framingham heart study, obesity as defined by body mass index (BMI) ≥ 30 was associated with incident “unprovoked” VTE (HR 2.74, 95%CI, 1.75–4.30) [64].

Gregson et al. analysed the relationship of different markers of adiposity (BMI, waist–to–hip ratio and waist circumference) with VTE. The HR for “unprovoked” VTE were 1.40 (1.34–1.46) per 1-SD higher BMI, 1.58 (1.43–1.75) for waist–to–hip ratio and 1.67 (1.58–1.78) for waist circumference [57].

Obesity may cause thrombosis due to the activity of adipocytokines, such as leptin and adiponectin, increasing coagulation activity and inflammation and decreasing the fibrinolytic cascade [65]. Thus, leptin increases the expression of Plasminogen Activator Inhibitor-1 (PAI-1) in endothelial cells, impairing fibrinolysis and thrombus resolution [66].

There is also evidence for an increased release of IL-6 and leptin from adipose cells [67], which have a proinflammatory effect that, in combination with the blood stasis caused by the mechanical impairment of obese people, promotes the thrombus formation. In addition, increased leptin levels in obese patients seem to increase PLTs adhesion to fibrinogen and binding of soluble fibrinogen [66]. In addition, leptin may promote the generation of active TF.

Adiponectin is the most represented adipose-related cytokine, and its serum levels seem to be reduced in obese patients. Adiponectin had an anti-inflammatory, anti-atherosclerotic and antithrombotic effect by modulating endothelial function [66], smooth muscle cells, PLTs and macrophages [68].

### 3.5. Factor V Leiden (FVL)

Under physiological conditions, factor V is a procoagulant protein that accelerates the conversion of prothrombin to thrombin and is inactivated by activated protein C. Factor V Leiden (FVL) is a gain-of-function mutation causing resistance to activated protein C, which determines the hypercoagulable state and an increased risk of VTE [69]. FVL is the most prevalent inherited thrombophilia in the general population, but is rarely found in non-Caucasian populations. The mutation can be inherited in heterozygosis, with an estimated prevalence of 4.7% among Europeans or those with European ancestry [70], or in homozygosis, with a prevalence estimated at 0.06–0.25% [71]. The prevalence of this inherited thrombophilia increases to 19% in patients diagnosed with DVT [72] and to 28.4% in those with recurrent VTE [73]. The prevalence of FVL has been also investigated in patients with venous thrombosis in unusual sites. A metanalysis including 1822 cases of cerebral vein thrombosis (CVT) and 7795 controls showed an increased prevalence of FVL carriers in patients with CVT (OR 2.70, 95%%CI 2.16–3.38), but the association varies depending on the geographic area [74]. A meta-analysis including 1748 cases of retinal vein thrombosis and 2716 controls also showed an increased prevalence of FVL mutation in patients with retinal vein obstruction (OR 1.66, 95%CI, 1.19–2.32) [75]. An increased prevalence of FVL mutation was also found in a splanchnic veins thrombosis cohort, with a prevalence from 4 to 26% in patients with Budd–Chiari syndrome [76].

FVL mutation is associated with an increased risk of VTE. The strength of this risk factor is different depending on it being inherited in homozygous or heterozygous form. The risk of a first episode of VTE in a heterozygous FVL is about 3 times higher than in patients without (OR 3.61, 95%CI, 2.02–5.95) [77], while the risk of developing a first VTE in an homozygous carrier of FVL is about 11 times higher (OR 11.45, 95%CI, 6.79–19.29) [78]. The risk of recurrence in patients not receiving a proper anticoagulant therapy is also markedly increased (OR 5.81, 95%CI, 4.03–8.38) [78].

### 3.6. Prothrombin G20210A Gene Mutation (PGM)

Prothrombin G20210A Gene Mutation (PGM) is estimated to be the second most commonly inherited thrombophilia after FVL and was first described in 1996 by Poort and colleagues [79]. The G20210A mutation in the prothrombin gene is a substitution of guanine to adenine at position 20,210 in the 3-untranslated region of the coagulation factor II gene [79]. This gain-of-function mutation results in a higher level of prothrombin, and so also, the level of thrombin may rise, resulting in an increased risk of VTE [80].

The prevalence of PGM is 2.0% in the general population [81] and ranges between 3–22% in patients with a portal vein thrombosis [76]. An increased risk of having PGM was also reported in patients with a cerebral vein thrombosis (OR 5.84, 95%CI, 3.96–8.58) [82].

The risk of developing a first episode of VTE in PGM carriers is increased by nearly 3-fold (OR 2.80, 95%CI, 2.25–3.48) [78], while the risk of developing a portal vein thrombosis is estimated to be 4 times higher (OR 4.48, 95%CI, 3.10–6.48) than in non-carriers.

While the prevalence of PGM in patients who develop a recurrence in VTE is estimated to be 6% [72], PGM carriers do not have an apparent higher risk of VTE recurrence (OR 1.45, 95%CI, 0.96–2.21) [83], challenging the need for long-term anticoagulation in these patients.

### 3.7. Plasminogen Activator Inhibitor-1

PAI-1 is a serine protease that plays a key role in the fibrinolytic system by inhibiting fibrinolytic activators tPA and uPa. Through this mechanism, PAI-1 compromises the conversion of plasminogen to plasmin, which dissolves fibrin blood clots [84]. The 4G/4G homozygous genotype of PAI-1 is a gain of function and is associated with a higher level of PAI-1 that may compromise fibrin clearance with a pathological fibrin deposition and an increased risk of VTE [85].

In a study by Folsom et al., the prevalence of this mutation, in its homozygous form, is estimated to be 7.2% in patients presenting with an episode of VTE and 3.4% in healthy controls [86].

PAI-1 mutation is associated with an increased susceptibility to VTE (OR 1.25, 95%CI, 1.05–1.49) when considered in its homozygous form 4G/4G versus the wild type 5G/5G, and with an OR of 1.38 (95%CI, 1.06–1.81) when considering overall the carriage of a mutated allele versus the wild type [85]. Data on the incidence of recurrent VTE are still lacking.

### 3.8. Oral Contraceptives and Hormonal Replacement

Combined oral contraceptives and hormonal replacement therapy are established risk factors for thrombosis, both arterial and venous, with a higher incidence for the latter. Millions of women worldwide are estimated to be under this treatment, and this therapy is generally preceded by an evaluation of the patient’s thrombotic risk [87]. Hormonal treatment is usually composed of an association of oestrogens and progesterone.

The use of combined oral contraceptives carries an increased risk of a first VTE episode, with a RR of 3.5 (95%CI, 2.9–4.3), although the risk may increase with higher oestrogen doses (>30 mcg of ethinyl oestradiol) and non-levonorgestrel progestin [88]. Hormonal replacement therapy is associated with a milder risk of VTE (OR 2.35, 95%CI, 1.9–2.9), and this risk is even lower when patients are under non-oral hormonal replacement therapy. The risk of VTE in oral vs. non-oral intake is 1.8 times higher (OR 1.8, 95%CI 1.35–2.29) [89].

A metanalysis also showed an association between hormonal contraceptive therapy and an increased risk of CVT, with an OR of 7.95 (95%CI, 3.82–15.02) compared to untreated women [90].

Being a moderate intensity risk factor, compared with non-users, the OR for recurrence among estrogen-containing-contraceptive users was 0.4 (OR, 95%CI, 0.2–0.8), indicating a very safe profile in terms of recurrence [91].

As mentioned, hormonal therapy is linked to thrombotic risk mainly by oestrogens dose and oral administration, although the mechanisms are not fully understood [87]. This kind of treatment has been shown to increase the generation of thrombin, with a raise of D-dimer level and thrombin generation (F1 + 2) [92,93]. In addition, hormone therapy also plays a role on the regulation of endothelial function. A few reports suggest a dose-dependent role of oestrogens in expressing matrix metalloproteinases, which damage intima’s collagen and elastin, determining venous stasis, increased vascular permeability and thus facilitating venous thrombosis [94].

### 3.9. Cancer

Cancer represents a strong risk factor for the development of cancer-associated thrombosis (CAT), with an estimated increased risk of VTE 4 to 6.5 times higher than in people without cancer [95].

The prevalence of CAT is growing due to the longer patient survival; to a wider cancer surveillance, which increases the early and major detection of silent thrombosis; and to the major use of central venous catheters (CVC). Indeed, CAT represents almost 20% of the overall incidence of total VTE [96,97]. In the RIETE Registry, active or actively treated cancer accounts for 17% (n = 6075) of 35,359 patients with VTE [98].

The risk of VTE is highest in the first 3 months after cancer diagnosis (OR: 53.5; 95%CI: 8–334.4) [99], probably related to cancer treatments (surgery, chemotherapy or radiotherapy). Furthermore VTE, especially in atypical sites [100], is often the first and only sign that leads to diagnosing occult malignancy.

On top of that, CAT plays an important role in defining the mortality and morbidity of these patients; it represents the second leading cause of death after disease progression [101].

The risk of CAT varies according to tumour-related factors, such as cancer site and stage, malignancy treatment, patient related factors and biomarkers. A metanalysis including 57,591 patients, representative of 8 types of cancer, showed an overall risk of VTE of 13 per 1000 person-years, increasing to 68 per 1000 person-years in patients with metastatic disease [97]. The highest risk was seen in pancreatic cancers (110/1000 patient-years), while it was much lower in breast and prostate cancer (10/1000 patient-years) [97].

In addition, cancer treatments may favour CAT onset [100]. The use of thalidomide [102], lenalidomide [103] and hormonal therapy is associated with an increased risk of venous and arterial thrombosis (RR 1.6; 95%CI: 1.3–2.1) [99]. It is not well known yet if a role can be played by radiation therapy or new targeted therapies [104]. Among these, there are some studies about antiangiogenic agents, such as bevacizumab [105], which has showed a higher risk of CAT in patients treated with itself and cisplatin compared to patients with non-cisplatin or non-bevacizumab therapies (1.67, 95%CI, 1.25–2.23 and 1.33 95%CI, 1.13–1.56 respectively) [106,107].

In the Computerized Registry of Patients with Venous Thromboembolism (RIETE) study, recurrent CAT at 3 months occurred in 11.4% of patients with cancer vs. 2.1% in those without (*p* < 0.001) [108].

In a prospective study, the 1-year rate of CAT was high in gastrointestinal (HR 5.1 95%CI: 2.3–11.3) and lung cancer (HR 6.9 95%CI: 3–15.9) [109].

The risk of CAT/VTE according to cancer type is reported in Table 4.

The association between cancer and VTE is based on complex mechanisms that result in a hypercoagulative state [111], determined by the massive release of inflammatory cytokines, by the expression of haemostatic proteins on tumour cells and by the activation of the clotting system [112]. Non-haematologic cancers determine the release of granulocyte stimulatory factors, which leads to an important leukocytosis [113], especially neutrophils filled with NETs. These granules are put out in response to the inflammatory tumour microenvironment, in addition to the thrombocytosis [114,115] of cancer patients, the direct contact-pathway of coagulation cascade. Indeed, elevated leukocyte [116,117] and platelet counts have been associated to an increased risk of VTE in patients with cancer. One study reported that patients with platelet count >443 × 10^9^/L have a risk of CAT 3.5-fold higher than patients without malignancy [114].

Massive release of TF is another characteristic of many tumours; it is provoked by the presence of microparticles (MPs), cellular fragments of platelets or endothelial cells of 0.1–1 µm diameter, found in pancreatic, brain and lung cancer, especially in patients with high severity or metastatic disease. TF and MPs and P-selectin, which can be found on activated platelets or endothelial cells and whose levels above 75% percentile increase of 2.5-times the risk of VTE, are not only responsible for the activation of coagulation cascade, but also avoid immune detection of tumour cells and disease progression [118]. Chemotherapy and placement of long-term CVC determine a further risk factor of thrombosis, damaging the endothelial surface and releasing NETs, which promote clot formation.

The risk of CAT may be increased by some specific patient’s characteristics, such as obesity, age (>65 years), medical comorbidities and ethnicity [119,120]. In a study of White et al., VTE incidence was significantly higher in Caucasians than in Hispanics and Asian patients [121].

### 3.10. Long-Haul Flight

The first cases of VTE associated with air travel were reported in 1954. Since them, other cases have been described, also regarding atypical thrombosis [122]. The overall VTE risk may be as high as 1.2% if we depend on ultrasound screening studies, but it is around 0.05% counting only symptomatic DVTs [123], and the relative risk varies according to controls selection criteria, ranging from 1 with controls referred for VTE evaluation to 3 with non-referred control participants [124,125,126]. However, the association seems to be stronger with the increase of travel duration [124,125]. The WHO Research into global hazards of travel (WRIGHT) project concluded that the absolute risk of VTE per more than 4 h flight, in a cohort of healthy individuals, was 1 in 6000 [127]. In a case control study with 210 patients with VTE and 210 healthy controls, the relative risk of VTE was higher (3-fold) when only long-distance flights were considered [128]. Parkin et al. performed an observational study on 121 men and women with fatal PE, 11 of them with a history of long-haul flight in the past 4 weeks, and concluded that 1.3 (95%CI 0.4–3.0) for a million passengers with a flight > 8 h developed PE [129]. In a case-control study with 88 fatal PE cases and 334 healthy controls, the same investigators found that the adjusted OR for travellers who had flown for >8 h was 7.9 (95%CI, 1.1–55.1) [129]. According to MacCallum et al., the risk associated with air travel is not limited only to long individual flights, but also to cumulative flight time. In their study with 550 VTE cases and 1971 controls, cases were two- to three-fold more likely than controls to have flown >4 h in any one leg of their journey or >12 h in total over the previous 4 weeks [130].

However, VTE risk associated with long-haul-flight increases with the presence of other VTE risk factors, such as a previous thrombosis history. In the above research from MacCallum et al., cases were 8-fold more likely to have a previous history of VTE than controls [130].

The increased risk of VTE seems to be connected to two different factors, such as immobilization and hypobaric hypoxia. Immobilization and cramped seating during long-haul flights carry a major risk of VTE due to the compression of popliteal veins [126], with blood stasis and activation of the coagulation cascade. On the other hand, hypobaric hypoxia is responsible for the inhibition of the fibrinolytic system and the generation of thrombin, as shown by increased levels of D-Dimer and reduced levels of PAI-1 in travellers [131].

### 3.11. Antiphospholipid Syndrome (APS)

Antiphospholipid syndrome (APS) is an immune-mediated disease characterized by thrombotic and/or obstetrical events [132]. Thrombotic APS is characterized by venous, arterial or microcirculation thrombosis [133]. This disease is caused by antibodies directed against membrane anionic phospholipids (mainly anticardiolipin and anti-phosphatidylserine antibodies) or their associated plasma proteins, mainly beta-2 glycoprotein I (β2GPI), or the presence of a lupus anticoagulant (LAC) [133,134].

VTE in APS patients usually occurs as DVT of lower limbs [135], but venous thrombosis in unusual locations, such as hepatic veins or cerebral venous circulation, are also common, while arterial thrombosis generally affects the cerebral arterial circulation [135]. The exact thrombotic risk associated with this condition needs to be completely investigated.

The antiphospholipid antibodies’ profile shows a strong association with the risk of thrombosis. LA positivity carries the strongest association with both arterial and venous thrombosis [136]. A positivity for LAC and anti- β2GPI antibodies drastically increased the risk of thrombosis (OR 4.1, 95%CI: 1.3–13.5) [137]. The risk of a first VTE among asymptomatic subjects who are positive for LAC, anticardiolipin and anti–β2GPI antibodies (triple positive phenotype) is 5.3% per year. Furthermore, if an anticoagulant therapy is not implemented, up to 44% of triple positive APS patients will undergo recurrent thrombosis over a 10-year follow-up period [138]. In some patients with clinical features of APS, VTE may occur in the absence of the positivity of classical antiphospholipid antibodies (aPL), and these patients are referred to as “seronegative APS” [139].

The mechanisms through which aPLs induce thrombosis are unclear. First, there are the interactions with the coagulation and fibrinolytic systems, particularly inhibition of the protein C system, but also the interactions with antithrombin, TF and tissue-type plasminogen activator. In addition, there are other factors, such as vascular cells activation like endothelial cells, monocytes, neutrophils and platelet by aPL, particularly anti-β2GPI, complement activation and the destruction of annexin V and exposure of the procoagulant phosphatidylserine on the cellular surface [140,141].

### 3.12. Residual Venous Thrombosis (RVT)

An emerging risk factor for VTE recurrence is represented by residual venous thrombosis (RVT). According to a systematic review, RVT found within three months after acute event of thromboembolism appeared to be associated positively to a major risk of recurrence (OR 2.02, 95%CI, 1.62–2.50) [142].

Another cohort study [143], which included 55 patients with history of VTE in the past 33 months, used ultrasound examination to study the echogenicity of residual thrombosis as a marker of recurrence.

It was confirmed that ultrasound sonography allowed predicting recurrence in 75% of cases and suggested hypoechogenic thrombi (Gray Scale Medians [GSM] < 24) as a predictive marker of TVP recurrence [143].

As it that were enough, a review [144] that considered a cohort of 313 consecutive symptomatic outpatients with proximal venous thrombosis who had a standard anticoagulation showed a hazard ratio of recurrent thromboembolism of 2.4 (95%CI, 1.3 to 4.4; *p* = 0.004) for persistent residual thrombosis versus veins considered recanalized using ultrasound echography.

As a matter of fact, it has been supposed that the damaged endothelial wall and the blood stasis connected to the residual clot promote thrombotic recurrence [145], but it has been observed that new thrombotic events develop also in the unaffected leg or as isolated PE. Other pathophysiological mechanisms need to be explored.

### 3.13. SARS-CoV2 Disease (COVID-19)

Since the start of the pandemic outbreak, an increased risk of VTE during SARS-CoV-2 infection has been reported in several studies of different design, size and quality, but there has been a high variability of reported rates. The large variability may be related to differences in diagnostic protocols or screening for VTE, such as outpatients vs. hospitalised and non-intensive care unit (ICU) or ICU setting or use of antithrombotic prophylaxis. Hospitalized patients with COVID-19 have many common risk factors for VTE as other inpatient, but severe SARS-CoV-2 results in an increased risk of thrombotic complications, occurring both in the venous and arterial system [146,147].

For example, a meta-analysis by Nopp et al. that involved 28,173 patients (1819 clinic, 20,886 non-intensive care unit [ICU] hospitalized and 5468 ICU patients) reported an overall VTE prevalence of 14.1%, 40.3% with ultrasound screening and 9.5% without screening. Subgroup analysis revealed high heterogeneity, with a VTE prevalence of 7.9% (95%CI, 5.1–11.2) in non-ICU and 22.7% (95%CI, 18.1–27.6) in ICU patients [148]. Moreover, Hasan et al. in 2020 in a metanalysis that included twelve studies reported a VTE prevalence of 31% among ICU patients, despite the use of prophylactic or therapeutic anticoagulation [149].

As a matter of fact, the occurrence of VTE increases the risk of death both in outpatients (HR 4.42, 95%CI, 3.07–6.36) and inpatients (HR 1.63, 95%CI, 1.39–1.90) [147].

Moreover, many studies showed that the risk of hospital-associated VTE extends from the time of admission and within the first 90 days post hospital discharge also in COVID-19 patients [150,151]. These findings suggest that post discharge anticoagulation therapy may be considered for high-risk patients with COVID-19, such as those with a history of VTE, D-dimer > 3 μg/mL and predischarge C-reactive protein >10 mg/dL [152,153]. Recent data showed that risk of post-discharge VTE seems to be reduced across COVID-19 waves, decreasing from 3% in the pre-Delta, to 1.7% in the Delta and 0.9% in the Omicron wave [154]. This decrease may be due to the extensive use of anticoagulation and to the introduction of vaccination. In this regard, the risk of VTE has been modified by the introduction of vaccination. The rate of VTE in 2020 (pre-vaccination was estimated at 21% (95%CI: 17–26%), raising to 31% (95%CI: 23–39%) in ICU patients [155].

A population study including 18,818 outpatients showed that VTE risk decreased from an HR 21.42 to 5.95 in the fully vaccinated participants who were then infected by SARS-CoV2 [156]. A study performed in the emergency department showed a nearly 3-fold increase in the risk of PE in unvaccinated patients (HR 2.75, 95%CI, 1.14–6.73) [157].

COVID-19 is associated with coagulopathy favouring VTE through several mechanisms. First, SARS-CoV2 interacts with the angiotensin converting enzyme (ACE)-2 receptor on endothelial cells, which results in an increased release of the vasoconstrictor angiotensin-II and an endothelial dysfunction [158]. In addition, inflammatory response plays a primary role through complement activation, elevating levels of proinflammatory cytokines, such as interleukin-6 (IL-6) and IL-17A, which activate platelets, tissue factor and then the coagulation cascade [159,160]. Moreover, recent studies have shown alterations of both coagulation and fibrinolysis by multiple pathways, such as reduction of antithrombin and protein C and increasing of PAI-1 [160,161]. This hypercoagulation [162] and cytokine storm result in the alveolar thrombosis of SARS-CoV2 disease.

### 3.14. Trauma and Fractures

Fractures are a strong provoking factor for VTE. In a study of 480 patients, the incidence of fractures in the 90 days before hospitalization for VTE was 11.8%, compared with incidence during the control period that was 3.6%, with an adjusted incidence rate ratio of 2.81 (95%CI, 1.57–5.03) [163]. Particularly in lower limb injuries, VTEs are a common complication, with incidence that varies among different types of fracture (reported in Table 5). Of note, in all studies, the majority of DVTs were asymptomatic.

Interestingly, a retrospective study reported that in patients with hip fractures 38.89% of all preoperative DVTs were on the uninjured site, suggesting that not the local fracture itself, but a post-injury hypercoagulation state may be the main contributing factor to this phenomenon [172].

A particular setting is the risk of cerebral sinus-vein or jugular veins thrombosis after a cranial trauma. Among 90 patients with cerebral sinus-vein or jugular veins thrombosis, history of trauma appeared to be the main risk factor, with 14.4% of patients reporting a head trauma, with or without fracture, in the month preceding the DVT event [173].

Fractures play a role in developing VTE because of the inflammatory storm consequent to the trauma and especially because of high levels of TF. As a matter of fact, trauma always represents a physical disruption of the endothelium, which leads to hypoxia and haemodynamic stress, involving in this phenomenon a large variety of cytokines (IL-1, IL-6, IL-8), chemokines, antibodies and immunocomplexes [174]. Damaged endothelial cells release into the bloodstream TF and phospholipids contained in microparticles, which are responsible for thrombus formation [175].

### 3.15. Trauma and Immobilization without Fractures

Lower limb traumas other than fracture are not associated with VTE risk. In a case-crossover study on a cohort of 480 patients with VTE, the adjusted IRR for VTE for open wounds was 0.74 (95%CI, 0.32–1.73), for sprain 1.30 (95%CI, 0.59–2.91) and for dislocation 1.33 (95%CI, 0.34–5.24) [163]. However, lower limbs immobilization after a trauma is a risk for VTE itself, irrespectively of the presence of fractures. A systematic review with 15 studies and 80,678 patients with temporary lower limb immobilization due to an isolated trauma showed that the prevalence of VTE from the studies was 4.8% (0.22% to 23.5%) [176].

Immobilization by itself, complicated or not by trauma, has a direct effect on coagulative cascade because it determines blood stasis. To this, cellular margination and local hypoxia follow, and so endothelial activation, which is the beginning of the thrombotic process. By the way, data from animals have shown that blood stasis is not enough to activate coagulation [177]; TFs contribute, probably released or expressed on activated endothelial cells or mononuclear cells, called by cytokines (IL-1, IL-6 and IL-8) produced in the inflammatory microenvironment due to blood stasis.

### 3.16. Major and Minor Surgery

**Major surgery.** In 2020, a 3-round Delphi process consensus defined major surgery according to pre-existing comorbidity of the patients, extent and complexity of the procedure (as intraoperative blood loss >1000 mL; high vasopressor dose, vascular clampage or organ ischemia, long operative time), its pathophysiological consequences (as 30-day overall morbidity and mortality) and consecutive clinical outcomes (as systemic inflammatory response or need for intensive care) [178]. Therefore, although most surgical procedures increase risk for VTE, this varies considerably across surgical procures and among individual patients undergoing surgery. Current guidelines suggest considering at high VTE risk surgeries with >45 min under general anaesthesia [179]. Surgical procedures carrying the highest risk of postoperative VTE include hip and knee arthroplasty [180]. In non-orthopaedic surgery, open abdominal and open pelvic procedures are associated with a high risk of VTE [180]. VTE risk appears to be highest for patients undergoing abdominal or pelvic surgery for cancer [180]. In addition, independent risk factors of the patient affect the cumulative risk of VTE.

**Minor surgery.** VTE events appear to be significantly lower in laparoscopic surgery when compared with open surgery. In a retrospective study that analysed data of 138,595 patients obtained from the University HealthSystem Consortium Clinical Database, the incidence of VTE among patients who underwent laparoscopic surgery was 0.28% versus 0.59% in patients who underwent open surgery (OR 1.8, 95%CI, 1.3–2.5, *p* < 0.01) [181]. Another study analysed data of 750,159 patients from The National Surgical Quality Improvement Program database who underwent abdominal laparoscopic surgery. The incidence of VTE was 0.32% within 30 days of operation, with the highest incidence among patients who underwent colorectal surgery (1.12%). In this study, the length of hospitalization and the duration of surgery appeared to be significantly longer in patients with VTE [182].

Even other retrospective studies found a relationship between time of surgery with general anaesthesia and VTE risk [183,184,185]. Another factor associated with different risk for VTE was patient positioning during the surgery. In a retrospective study on 374,017 subjects undergoing laparoscopic surgery, patients were divided into three subgroups based on positioning during the surgery: Trendelenburg position, reverse- Trendelenburg and supine. The lowest VTE risk was observed in reverse-Trendelemburg, compared with supine (OR 1.49, 95%CI, 1.16–1.93) and Trendelenburg (OR 1.34, 95%CI 1.15, 1.56) positions [185].

Surgery is a transient condition associated with an increased risk of VTE. Different mechanisms are involved, but venous stasis, endothelial activation and local accumulation of TF are the main ones responsible for the development of VTE [80].

Venous stasis occurs both during and after surgery, especially in major orthopaedic surgery [186], and it is responsible for increased haemostasis, cellular margination and local hypoxia, which amplifies endothelial activation [80]. In addition, TF plays a fundamental role during haemostasis and is stimulated by disruption of the endothelium during surgery and by inflammatory mediators (including cytokines, chemokines, VEGF and factors derived by complement activation), which are increased post-operatively [177,187].

### 3.17. Pregnancy

Pregnancy is considered a transient risk factor for VTE. A Dutch registry including 1,919,918 women showed an incidence rate of VTE of 2.3 per 10,000 person-years [188]. The rate of VTE may be lower, taking into account the use of prophylactic low-molecular weight heparin during pregnancy [189]. A meta-analysis including >93 million pregnant and postpartum women showed an incidence of VTE of 1.2 per 1000 deliveries [190].

A recent Korean study showed an incidence of pregnancy-related VTE of 2.62/10,000 deliveries, this risk increasing from 1.80 for women aged 20 years to 5.46 for women in their 40s (relative risk, 3.03; 95%CI, 2.04–4.51; *p* < 0.01) [191]. Women using assisted reproductive technology (ART) seem to have a two- to threefold increased risk of VTE as compared to spontaneous pregnancy (relative risk [RR]: 2.66; 95%CI: 1.60–4.43) [192].

Some factors, such as hypertension occurring during the pregnancy, seem to be implicated in VTE risk. Indeed, VTE risk is higher in women with hypertension (HR 2.0, 95%CI, 1.7–2.4), and highest among women with preeclampsia (HR 7.8, 95%CI, 5.4–11.3) compared to those without [188].

The mechanism underlying the risk of VTE during pregnancy may rely on the physiological reduction of some natural anticoagulant proteins, such as protein S, that may be reduced by 40–50% of normal levels. However, it remains to be established whether this reduction leads to an increased risk of VTE [193]. Similarly, antithrombin levels may be reduced by 20% during pregnancy [194].

## 4. Conclusions

The incidence of VTE is variable across countries and seems to be lower in Eastern Countries. Intrinsic and environmental risk factors may lead to VTE through several different mechanisms that are frequently overlapping (Figure 2). The exact role of some of these risk factors and their combination needs to be further investigated. The search for new risk factors for VTE continues given the still high rate of idiopathic VTE, as well as the improvement of risk stratification strategies to prevent recurrent VTE.

## Figures and Tables

**Figure 1 ijms-24-03169-f001:**
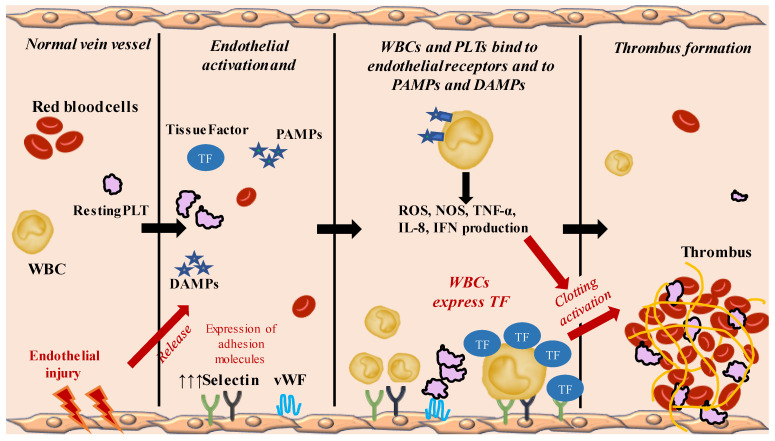
Summary of general mechanisms of venous thrombosis.

**Figure 2 ijms-24-03169-f002:**
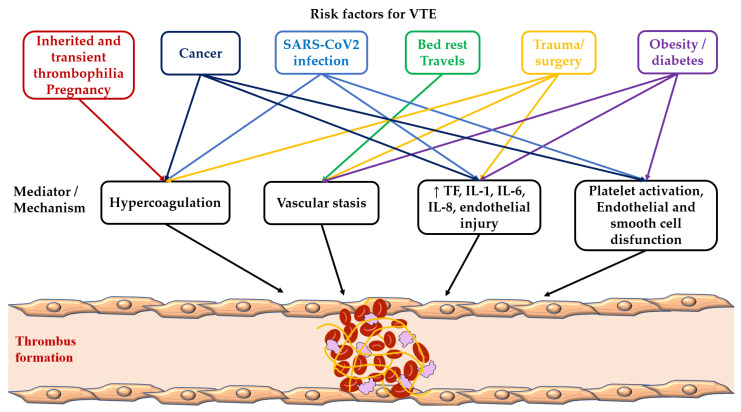
Risk factors and mediators of venous thromboembolism.

**Table 1 ijms-24-03169-t001:** Incidence (per 1000 person-years) of venous thromboembolism according to different countries.

Country	Incidence of VTE	Incidence of DVT	Incidence of PE
Norway [2]	1.43	0.93	0.50
Italy [3]	-	-	0.41–0.55
France [4]	1.84	1.198	0.64
Spain	1.54 [5]	-	0.33 [6]
Germany [7]	1.4–3.2	1.1–2.9	0.8–1.6
United Kingdom [8,9]	0.75–1.31	0.40	0.34
Denmark [10]	1.15	0.65	0.51
United States of America [11]	1.17	0.48	0.69
Canada [12,13]	1.22–1.38	0.78	0.45
Australia [8]	0.83	0.52	0.31
Taiwan [8]	0.16	-	-
Hong Kong [8]	0.08	0.17	0.04
Korea [8]	0.14	0.05	0.07
Argentina [8]	1.65	1.30	0.69

DVT: Deep Vein Thrombosis, PE: Pulmonary Embolism, VTE: Venous Thromboembolism.

**Table 2 ijms-24-03169-t002:** Risk factors for venous thromboembolism.

**Weak Risk Factors**
Bed rest >3 days/prolonged travel Cardiovascular risk factors (Diabetes mellitus/Arterial hypertension/Obesity) Elderly Minor surgery Pregnancy/puerperium Varicose veins
**Moderate Risk Factors**
Arthroscopic knee surgery Autoimmune diseases (Sjogren’s syndrome, rheumatoid arthritis, systemic lupus erythematosus, vasculitis and systemic sclerosis) Blood transfusion/Erythropoiesis-stimulating agents Central venous lines/Intravenous catheters and leads Chronic congestive heart failure or respiratory failure Hormone replacement therapy/In vitro fertilization/Oral contraceptive therapy Infection (specifically pneumonia, urinary tract infection and HIV) Inflammatory bowel disease Cancer (highest risk in metastatic disease)/Chemotherapy Paralytic stroke Superficial vein thrombosis Thrombophilia
**Strong Risk Factors**
Fracture of lower limb/Hip or knee replacement/Spinal cord injury Hospitalization for heart failure or atrial fibrillation/flutter or Myocardial infarction (within previous 3 months) Major trauma Previous VTE Antiphospholipid syndrome Thrombophilia (homozygous of Factor V of Leiden or prothrombin 20210, antithrombin deficiency and combination thrombophilia)
**Uncertain Risk Factors**
Thrombophilia (heterozygous of Factor V of Leiden or prothrombin 20210, PAI-1 mutation and Protein C and S deficiency) Male sex Smoking habits Myopathies

PAI-1: plasminogen activator inhibitor-1, VTE: venous thromboembolism.

**Table 3 ijms-24-03169-t003:** Pathogenesis of risk factors for VTE.

Risk Factor	Stasis	Vascular Disfunction/Injury	Abnormal Coagulation Cascade	Innate Immunity Activation/Increase of Inflammatory Mediators	Higher Number of Platelets
** *Aging* **	-	+	+	+	-
** *Venous catheter insertion* **	-	+	-	-	-
** *Hormonal therapy/oral contraceptives* **	-	-	+	-	-
** *Trauma* **	+	+	-	+	-
** *Surgery* **	+	-	-	+	-
** *Prolonged bed rest/plaster cast* **	+	-	-	-	-
** *Long-haul flight* **	+	-	-	-	-
** *Diabetes* **	-	+	+	-	-
** *Obesity* **	+	-	+	-	+
** *Smoking* **	-	-	+	+	+
** *SARS-CoV2 infection* **	-	+	-	+	-
** *Infection/sepsis* **	-	+	-	+ (also acquired immunity)	-
** *Inflammatory disease* **	-	-	-	+ (also acquired immunity)	-
** *Cancer* **	-	+	-	+	+
** *Chemotherapy* **	-	+	-	+	-
*Inherited and acquired thrombophilia*	-	-	+	-	-

**Table 4 ijms-24-03169-t004:** Risk of venous thrombosis according to cancer type [110].

Type of Cancer	Incidence Rate (per 1000 Person-Years)	Hazard Ratio (95%CI)
Pancreatic	156.0	50.4 (36.5–69.6)
Ovarian	71.8	30.7 (21.0–45.1)
Liver	103.6	23.1 (13.4–39.8)
Lymphoma - *Hodgkin* - *Non-Hodgkin*	60.5 59.5	95.8 (22.9–401.2) 20.1 (15.5–26.0)
Leukaemia	29.8	9.6 (7.1–13.0)
Stomach	66.4	20.1 (14.0–28.9)
Colon	51.9	12.8 (11.2–14.7)
Brain	54.6	23.0 (14.7–35.9)
Bladder	37.8	8.9 (6.9–11.4)
Kidney	51.2	22.3 (15.4–32.3)
Melanoma	7.3	2.9 (2.2–3.8)
Prostate	16.5	3.8 (3.3–4.3)
Lung - *Non-small cells* - *Small cells*	74.4 43	20.0 (17.4–22.9) 14.8 (10.0–21.9)
Breast	13.2	4.5 (3.9–5.1)
Uterine	30.1	10.8 (7.8–14.9)

CI: confidence interval.

**Table 5 ijms-24-03169-t005:** Incidence of VTE according to the type of fracture.

Type of Fracture	Incidence of VTE	Reference
**Hip**	DVT: 16.6%	[164]
**Pelvis**	DVT: 0.21–41% PE: 0–21.7%	[165]
**Tibial plateau**	DVT: 16.3% (86.4% of DVTs diagnosed within 7 days after the injury and 66.0% within 2 days)	[166]
**Patellar**	DVT: 4.4% DVT: 5.8%	[167,168]
**Calcanear**	DVT: 12% DVT: 3.1%	[169,170]
**Tibial shaft**	DVT: 13.3%	[171]

## Data Availability

Not applicable.

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
