# Peer review of "A Comprehensive Review of Risk Factors for Venous Thromboembolism: From Epidemiology to Pathophysiology"

_ijms, 2023, doi:10.3390/ijms24043169_

Round 1

Reviewer 1 Report

Well-written review article, clearly and comprehensively presents VTE risk factors and their clinical aspect. I believe that it is suitable for publication in its present form and in my opinion does not require any correction.

Author Response

Answer: Thank you for your kind comments. We are happy that you appreciated our work.

Reviewer 2 Report

Thank you for the invitation to review this manuscript. This draft provides an insight into risk factors along with clinical consideration for VTE. I have some concerns as given below;

Abstract: there is a need to provide the results in the abstract section. The abstract section is very limited with basic information. The authors are encouraged to enlist the major risk factors and provide directions for future research and clinical practice in the conclusion section.

The introduction section is divided into very small passages at the start. I would like to suggest the authors include the basic details of the VTE, its burden on healthcare system, prognosis and associated cost in the first paragraph of the manuscript. The second paragraph should emphasis the importance on understanding the pathophysiology of the VTE. For example, what will be the impact of prior knowledge on predictive scoring and risk factors of high-risk patients? How prior knowledge on the risk factors will alleviate the risks of VTE or improve the care of such patients.

I am not sure about the rational and importance of such a review, as there are many studies related to this topic in the literature. The authors did not provide any information that how this study covers the literature gap and improves the care of the patients. What are the other review questions that are not answered by previous reviews (as given below)

https://www.ahajournals.org/doi/full/10.1161/01.cir.0000078469.07362.e6https://www.dovepress.com/magnitudes-of-risk-factors-of-venous-thromboembolism-and-quality-of-an-peer-reviewed-fulltext-article-VHRM#:~:text=Advanced%20age%2C%20malignancy%2C%20trauma%2C,VTE%20reported%20from%20different%20studies.

Please clarify the definition of weak, moderate, severe or uncertain risk factors in the manuscript. 

Table 2: I suggest replacing the word mechanism of action with pathogenesis.

The authors did not provide any information on the funding, contribution, conflict of interests.

This manuscript seems to be written for the purpose of knowledge. However, the authors have also claimed to have clinical aspects in this manuscript. I hardly found a few clinical aspects in this manuscript. Again, I will emphasize the rationale of this review. The authors should define a clear review question and how this review is answering this question. This review can be further improved by incorporating clinical aspects of each risk factor. For example, how patient`s care can improve if they have non-modifiable risk factors, or modifiable risk factors. What will be the treatment considerations in the presence of each risk factor? The consideration may include selection of treatment, treatment duration and precautionary measures. I would suggest the authors maintain the two aspects of VTE in the manuscript. These aspects include pathophysiology of VTE along with its management in the presence of various risk factors, and in various age groups or special population.

Reviewer 3 Report

The manuscript conceived by Daniele Pastori et al. is well-written, comprehensive, but some aspects require to be further improved:

1.     In the Introduction section, you mentioned the epidemiology of DVT in Norway. Why specifically in this country? I highly suggest to include additional references concerning the epidemiology of DVT around the world, or at least in EU’s different regions (preferably summarized in a table/figure).

2.     The OR of cancer in developing DVT/VTE is an important issue and needs to be emphasized on. I suggest to systematize the cancer associated risk (expressed as OR) in a table, in order to be more visible.

3.     Pregnancy is an important aspect to be approached when it comes to DVT.

4.     SARS-CoV-2 is an already established risk factor for DVT. However, infections (and inflammation generally speaking) regardless of etiology are both directly and indirectly related to DVT (via Virchow’s triad elements). It would be interesting to provide additional data concerning some types of infection and their associated risk for DVT (e.g. severe bacterial pneumonia, chronic hepatitis, urinary tract infections, sepsis etc).

5.     Related to previous point: concerning the COVID-19 induced DVT, please provide some references concerning the incidence of DVT before and after the vaccination (consider 2020-no vaccination at all and after the start of the vaccination campaigns around the world/Europe)

6.     Minor observation: Simple Summary section is duplicated, please remove one (lines 572-578).

Best regards,

The Reviewer

Round 2

Reviewer 2 Report

Thank you for addressing the comments.